# Telomeric and Sub-Telomeric Structure and Implications in Fungal Opportunistic Pathogens

**DOI:** 10.3390/microorganisms9071405

**Published:** 2021-06-29

**Authors:** Raffaella Diotti, Michelle Esposito, Chang Hui Shen

**Affiliations:** 1Department of Biological Sciences, Bronx Community College, City University of New York, New York, NY 10453, USA; raffaella.diotti@bcc.cuny.edu; 2The Graduate Center, PhD Program in Biology, City University of New York, New York, NY 10016, USA; michelle.esposito@csi.cuny.edu; 3Department of Biology, College of Staten Island, City University of New York, New York, NY 10314, USA; 4The Graduate Center, PhD Program in Biochemistry, City University of New York, New York, NY 10016, USA; 5Institute for Macromolecular Assemblies, City University of New York, New York, NY 10031, USA

**Keywords:** telomere, sub-telomere, Aspergillus fumigatus, Candida albicans, Candida glabrata, Pneumocystis jirovecii

## Abstract

Telomeres are long non-coding regions found at the ends of eukaryotic linear chromosomes. Although they have traditionally been associated with the protection of linear DNA ends to avoid gene losses during each round of DNA replication, recent studies have demonstrated that the role of these sequences and their adjacent regions go beyond just protecting chromosomal ends. Regions nearby to telomeric sequences have now been identified as having increased variability in the form of duplications and rearrangements that result in new functional abilities and biodiversity. Furthermore, unique fungal telomeric and chromatin structures have now extended clinical capabilities and understanding of pathogenicity levels. In this review, telomere structure, as well as functional implications, will be examined in opportunistic fungal pathogens, including *Aspergillus fumigatus*, *Candida albicans*, *Candida glabrata*, and *Pneumocystis jirovecii*.

## 1. Introduction

One of the main global health concerns of the last few decades has been the increase in pathogen drug resistance. Although most of the focus has been on antibiotic resistance, in the last couple of decades fungi have emerged as some of the most devastating pathogens affecting human health [1].This underlies the need to better understand the process involved in fungal pathogenicity and virulence and to identify the strategies involved in the evolution of anti-fungal resistance [2].

Fungal infections are of particular importance as their targets are often subjects that have underlying health conditions or are immunocompromised and that are often found with other people in similar conditions, with nosocomial infections comprising most of the fungal infections outbreaks [1]. Many fungal infections are due to opportunistic organisms that grow commensally on skin and mucosal surfaces and are generally under the control of the immune system in healthy individuals, for example, pathogens belonging to the *Candida* spp. or *Aspergillus* spp. and *Pneumocystis jirovecii* (formerly *carinii*). Others belong to endemic species, such as *Coccidioides immitis* or *Blastomyces dermatitidis* that can affect both healthy and immunocompromised individuals.

The rise in incidence of these infections has spurred renewed interest in understanding the shift towards pathogenicity of these organisms, with the goal to improve prevention, diagnosis, and therapy. One area that has gained prominence in the study of many pathogens, including fungi, is the telomeric and sub-telomeric location of gene clusters containing genes known to be, or expected to be, involved in an increase of virulence and pathogenicity. Despite there being much conservation amongst various eukaryotes with respect to their telomeric and sub-telomeric composition, there also exists a great deal of diversity in the fungal telomeric repeats, telomerase RNA templates, and types/numbers of telomere binding proteins in telomeric complexes [3]. The understanding of these structures and their function has progressed significantly in the last few years due to the availability of better technical and analytical tools. Their role in many important processes in the cells have been discovered, including expression of genes critical for pathogens virulence and drug resistance [4]. These findings, revealing the proteins involved in the process of pathogenicity and their specificity to the organism could provide potential drug targets that would allow us to find new approaches to counteract the increase in antifungal resistance [5].

Telomeres are structures found at the end of linear chromosomes consisting of repetitive TG-rich DNA sequences and associated proteins (Table 1; [6]). Their maintenance is controlled by telomerase, which is an enzyme complex composed by a reverse transcriptase that catalyzes the extension of telomeres using an RNA template [7,8]. Their role is to maintain genomic integrity by protecting the chromosome ends from degradation due to replication and cell division. Therefore, telomeres are critical components of genomes as they maintain the integrity of unique sequences during replication and prevent chromosomal fusions [9].

In eukaryotes, telomere sequences generally consist of double-stranded (ds) region composed of 5′-TTAGGG-3′ tandem repeats that end with a single-stranded (ss) 3′ overhang (Figure 1; [3,10,11]) Telomere maintenance can also be achieved through alternative lengthening of telomeres (ALT) pathways [12]. Various protein complexes have been found to interact with the telomeres directly or indirectly [13,14]. In humans, the core protein complex is referred to as Shelterin and it is composed by six proteins: Trf1, Trf2, Rap1 [15], Tin2 [16], Tpp1 [17,18], and Pot1. Trf1 and Trf2 each bind the sequence 5′-YTAGGGTTR-3′ in duplex DNA, showing very low tolerance for single-base changes, while Pot1 has strong sequence specificity, binding single-stranded 5′-(T)TAGGGTTAG-3′ sites both at the 3′ end and at internal positions. Since these three shelterin subunits are connected through protein-protein interactions, shelterin has the capacity to recognize telomeric DNA with at least five DNA-binding domains (two each in Trf1 and Trf2 and one in Pot1) [13,19,20,21]. Furthermore, three other proteins, Ctc1, Stn1, and Ten1, forming the CST complex, binding single-stranded DNA at the telomeres have been identified [22]. As a consequence, shelterin is uniquely qualified to distinguish telomeres from all other DNA ends.

Shelterin-related complexes are also found at telomeres in other eukaryotes. Pot1-like proteins are present in nearly all eukaryotes, a Trf1/2-like protein is found in fission yeast and in trypanosomes. Telomeric proteins have been identified and characterized in yeast, such as fission yeast’s Rap1, which is a homologue of the mammalian protein, or Tpz1, Taz1, and Pot1, which are orthologs of the mammalian shelterin proteins Tpp1, Trf1/2, and Pot1. The role of these proteins encompasses regulation of telomere length and chromosome end protection, but most important for specific gene expression, they have a role in gene silencing. In *Saccharomyces cerevisiae*, for example, Rap1 recruits both the Sir complex (Sir 1, 2, and3) and Rif1 and Rif2. While Rif1 and Rif2 have a role in telomere length, the Sir complex is critical for gene silencing at telomeres and adjacent subtelomeric regions.

Gene silencing is a critical mechanism in fungal cells with subtelomeric regions adjacent to telomere being frequent targets of transcriptional repression via epigenetic regulators, such as histone deacetylases and methyltransferases [23]. Methyltransferases not only serve as recruitment and recognition sites for further repressive proteins, but also inhibit acetylation to promote a gene-silencing cascade [23]. Even though gene silencing may sound alarming, it has actually been found to be an invaluable mechanism at telomeric and subtelomeric regions, as this silencing appears to help maintain telomeric chromatin stability and prevents telomere-dependent cellular senescence through the prevention of decreased telomere lengths [23].

Sub-telomeric regions are also of great interest as pathogens can exploit the telomere position effect (TPE), an event that involves the silencing of genes located near the telomeric chromatin, to regulate gene expression. For example, TPE regulates the expression of secondary metabolites, cell surface proteins, such as adhesins, and other components that can support the organisms during the transition to pathogen. Many of these putative genes, or gene clusters, found at sub-telomeric regions on the chromosomes have been shown to be under epigenetic regulation, which may allow the organism to tailor its behavior to specific growth conditions [24]. Although telomeric instability due to issues with any of the telomeric proteins is generally correlated to the onset of cancer in humans and has a lethal effect on less complex organism, chromosomal rearrangements at the sub-telomeric regions may increase genomic plasticity and lead to increased virulence and drug resistance [25].

In this review, we discuss four of the most common organisms responsible for mycoses whose pathogenicity has been found linked to genes found at telomeric and sub-telomeric locations. Here, we will elucidate the role of the telomeric/sub-telomeric sequences in the clinical perspectives of these organisms.

## 2. Aspergillus fumigatus

*Aspergillus fumigatus* is one of the leading causes of mold-related health issues, mostly due to its propensity to produce large quantities of airborne spores that can easily infect human lungs. *A. fumigatus* is a ubiquitous saprophyte that can be both a primary and opportunistic pathogen in addition to being an allergen. Immune dysfunction, particularly a compromised immune system, permits the growth of inhaled spores in the human lungs and this causes aspergillosis, a significant threat to human health. The symptoms of aspergillosis can span from allergic reaction, such as asthma and sinusitis, to life-threatening invasive infections. The results of *Aspergillus* infections are determined by the condition of the host. In a healthy host, the spores can be easily cleared, while in immunocompromised individuals they can proceed to germination, colonization, and invasion of the pulmonary tissue. The mortality rate in infected immunocompromised individuals could reach up to 50% [26].

Telomeres in *A. fumigatus* are comprised of 7–21 tandem repeats of the common eukaryotic sequence TTAGGG (Table 1; [27]). Similar to the other fungi listed in this review, changes in the transcriptional expression of specific genes during infection affects the virulence and pathogenicity of *A. fumigatus.* Various candidate genes have been identified and the wide array of functions points to a complex picture. One hypothesis is that fungi secondary metabolites, which are generally organized in clusters, and tend to be species specific, could have a role in virulence. In *A. fumigatus,* there are 26 such clusters. A few of the clusters are unique to *A. fumigatus* and diverge from comparable family members, *A. oryzae* and *A. nidulans,* and others are missing. The location of these clusters of diverged and/or missing gene clusters shows a bias towards telomeric/subtelomeric locations [27] supporting the suggestion of greater genomic instability at these regions.

In an effort to identify the transcriptional profile of the saprophytic filamentous fungus during adaptation to the mammalian host niche and to shed light on the factors contributing to the pathogen virulence, McDonagh et al. [28] compared the transcriptome of developmentally matched *A. fumigatus* isolates from laboratory cultures and samples from murine lungs at initiation of infection. They identified factors, such as iron limitation, alkaline stress, and scarcity of nutrients as selective stresses at the beginning of pathogen infection. Interestingly, the analysis revealed that host adaptations genes affected during the early stages of the infection had a bias towards sub-telomeric locations [28]. Sixteen percent of *A. fumigatus* genes are predicted to be found 300 kb away from telomeres. Of the genes in that region, 29% of those sub-telomeric genes had an increase in expression in the murine samples at initiation of infection compared to the laboratory grown cultures. A few of the predicted gene clusters were uncharacterized; however, a good number of genes that belong to the fumitremorgin biosynthesis supercluster, located on chromosome 8 at 100 kb from the telomeres, were induced during early-stage infections, including the pseurotin biosynthetic cluster (Table 2). Pseurotin is a neuritogenic, nematicidal quinone. As such, it suggested that selective expression of a subset of secondary metabolite loci might facilitate initiation of mammalian infection [28].

LaeA is a methyltransferase known to be a global transcriptional regulator of secondary metabolites biosynthesis (Table 2; [30]). Loss of *laeA* (Δ*laeA*) results in reduced virulence in a murine model [33]. The transcriptional regulator influences the expression of ~9.5% of the *A. fumigatus* genome, and 20–40% of the secondary metabolites biosynthesis. Many of the secondary metabolites gene clusters under LaeA control are species-specific and strain-specific. The variability may be due to the fact that they are located at telomere proximal regions. In fact, 54% of the clusters differentially expressed in Δ*laeA* were found at a distance of ~300 kb from telomeres and showed a greater number of genes than other clusters located at other heterochromatin rich locations [33].

Comparison between the genes under LaeA control [33] and the ones expressed during early-stage infection in murine models [28] confirmed alignment of 99 genes out of 415 with 40% coding for secondary metabolites, including pseurotin, confirming the role of LaeA in *A. fumigatus* infection. Sub-telomeric genes were also preferentially expressed during nitrogen starvation indicating a role for sub-telomeric genes during nutrient starvation [28].

A recent paper investigating the shift of *A. fumigatus* to the biofilm phenotype identified a gene, *biofilm architecture factor A* (*bafA*), as one of the main players in the shift. *bafA* is part of a larger family of genes that can modulate the morphological changes needed to decrease the presence of vertically oriented hyphae and an increase in the ability of *A. fumigatus* to survive and thrive in a low oxygen environment, with a corresponding increase in the organism virulence. Similar to other genes involved in the organism pathogenicity, *bafA* is located within a subtelomeric gene cluster (Table 2; [31]).

Further studies will be needed to characterize the structure and function of gene clusters associated with virulence. New strategies, such as updated recombineering protocols [34] are being explored to overcome the technical difficulties involved in genetic manipulation of *A. fumigatus*. The sub-telomeric localization of many of the clusters suggests a role for epigenetic regulation of the genes, such as TPE (telomere position effect), which can effect gene expression [35]. It is conceivable that this strategy is used by the pathogen to tailor secondary metabolites synthesis to specific growth conditions. In addition, the diversity of genes found at telomeric and sub-telomeric regions may be due to the fact that they are hotspots for recombination events as it has been observed in many other organisms, for example, with chromosomal breaks followed by telomerase-mediated healing. In fact, in *ΔatmA* and *ΔatrA* mutants, some chromosomal polymorphism was observed suggesting their possible role in telomeric and subtelomeric maintenance (Table 2; [32]).

## 3. Candida albicans

*Candida* species (spp.) are an important component of the human microbiota and they have been cited as the fourth most common cause of nosocomial bloodstream infections [36]. Although pathogenic *Candida* spp. include *C. albicans*, *C. parapsilosis*, *C. tropicalis* and *C. glabrata*, *C. albicans* is by far the most commonly identified cause of candidiasis, and is often regarded as the most pathogenic fungal species in humans [37]. It normally resides within the vaginal mucosa as a harmless, lifelong normal flora of the microbiome, but under the right circumstances, it can cause opportunistic infections [38]. When conditions lead to robust overgrowth, when sexual contact transfers the fungus to a new location, or when a host is immunocompromised, *C. albicans* quickly becomes a nuisance, and can even progress to systemic complications, as it is known to have robust pathogenicity potential and adaptability [38]. Due to its unique life cycle quality and ability to switch from harmless normal flora microbiota to plaguing pathogen, *C. albicans* has become a point of interest in genome biology models and studies [39]. Recent studies have determined that the telomeric structure of *C. albicans* contributes to its pathogenicity and adaptability, while also contributing to the genetic biodiversity of these organisms [40,41]. 

Four major repetitive regions are present in the *C. albicans* genome: telomeres, sub-telomeres, the Major Repeat Sequence (*MRS*), and the rDNA locus (Figure 2; [40]). The telomeres of *C. albicans* have gained attention throughout the years for various reasons including their unusually long length [42]. These telomeres are composed of long tandem copies of a 23-bp repeat (5′-ACTTCTTGGTGTACGGATGTCTA-3′) that has considerably diverged from other *Candida* spp. with respect to both length and complexity [40]. Of the four fungal strains focused on in this review, *Candida*’s extended length becomes most obvious when compared to the standard TTAGGG sequence that is the most common telomere repeat in the animal and fungi kingdoms (Table 1; [43]). Normally, the repeat copy numbers of that 23-bp repeat are more than 20 copies per haploid genome and the size is about 500 bp to 5 kb.

A variety of significant regulatory regions in telomeres can influence pathogenicity and viability. Telomerase complex, which contains telomerase reverse transcriptase and telomerase RNA component, can maintain the length of telomere to prevent cell senescence via the production and maintenance of the 23 bp repeats at the telomeres [40]. Both telomerase RNA component and telomerase reverse transcriptase, encoded by *TER1* and *EST2* (ever shorter telomeres 2), respectively, perform repeated rounds of reverse transcription to build G overhangs as part of the end-protective T-loops that prevent telomere attrition in some fungal species (Table 3; [40,44]). More specifically, Est2 uses the *TER1* RNA as a template to add telomeric repeat subunits to the chromosome end in order to maintain replicative capacity and avoid cell senescence. The dual roles of end protection and telomere repeat addition by Est2 are separable. A catalytically inactive Est2 retains the ability to suppress excessive G-strand accumulation despite being unable to add telomeric repeats.

Within *C. albicans*, Est3 has been found to be an essential subunit of the telomerase regulatory complex with similarities to the telomeric protein, Tpp1, of mammalian cells (Table 3; [45,46]. It has been demonstrated that Est3 have a telomerase-stimulatory function required for telomerase activity [46]. Rap1 (repressor activator protein 1) is also critical for telomeres in *C. albicans* [47]. In many fungi, such as *S. cerevisiae*, Rap1 is required for cell viability. However, in *C. albicans*, it has been found that Rap1 is not required for viability, and instead functions as a critical regulator of telomeric length and structure [47]. *RAP1* is also conserved, with a human ortholog, *hRAP1* in existence [15]. Rap1 has been shown to have an unusually small size in *C. albicans*, and even lacks the usual C-terminal domain found in other fungal Rap1 proteins [47]. *C. albicans’* null mutants lacking *RAP1* are viable, but exhibit slowed growth and abnormal filament morphologies similar to those that would be observed in DNA repair mutants. These *rap1* null strains also exhibit unusually long telomeres with a complete lack of regulated length control. It has been shown that Rap1 works in conjunction with a series of other telomeric proteins, including Ku70, Stn1, and Ten1, to suppress abnormal recombination and maintain telomeres (Table 3; [47]). 

While the telomeric tandem repeats of those 23-bp remain identical amongst the strains of *C. albicans*, the surrounding sub-telomeric sequences have been found to be quite variable [42]. These sub-telomeric sequences have been found to be rich in transposable elements and are known hotspots for recombination events and mutations [40]. Genomic variations and the resulting phenotypic variations from those sub-telomeric variable sequences are crucial for biodiversity and evolution.

Amongst these variable sub-telomeric sequences, *TLO* genes (telomere-associated genes) have been expanded and are linked to the virulence and adaptability of *Candida albicans* as a pathogen (Table 3; [48]). Whereas other *Candida* species have one or two *TLO* genes, including the closely related *C. dubliniensis*, *C. albicans* has diverged to have a striking 14 to 15 *TLO* genes. In *C. albicans*, as the name suggests, the *TLO* genes are situated close to the telomeres of each chromosome. Three clades (α, β, and γ) of these expressed *TLO* genes have been identified. They all contain a Med2 domain, but the difference is based upon long terminal repeat insertions sequences [48]. Med2 is a component of the Mediator complex [41]. Mediator is a large multi-subunit protein complex, which is conserved throughout eukaryotes, and mediates interaction between RNA polymerase II, and the machinery used in the initiation of transcription at target gene promoters [41]. It has been shown that the deletion of the *TLO* regions in *C. dubliniensis*, a close relative to *C. albicans*, resulted in the loss of virulence through hindered hyphal growth and stress responses [41]. This is due to the defects in activation of transcriptional responses associated with a number of virulence traits including tolerance of oxidative stress and hypha formation. In *C. albicans*, hyphal growth, and increased adaptability are amongst the traits considered as contributing to its increased virulence compared to other *Candida* species [48]. It has been suggested that the expanded *TLO* regions with broad variability in expression levels and localization, can contribute to subsequent variability in the Mediator complex subunits and, thus, promotes increased adaptability and virulence [41,48].

Sub-telomeric rapid evolution is frequently attributed to their mutational hot spots and high recombination rates. These highly dynamic regions in *C. albicans* accumulated 31 point mutations, equivalent to one mutation per 515 base pairs, in the span of approximately 450 passages or 4500 cell divisions. Furthermore, duplication of *TLO* family genes and the loss of some *TLO* family genes were frequent results of recombination events observed at a rate of approximately one per 5000 divisions [49]. It was even noted that these quick rates were actually underestimates of the highly frequent recombination, as low frequency recombination events cannot be detected by the experimental equipment. Overall, the highly variable and dynamic recombination of Candida’s sub-telomeric regions helps drive evolution and promotes adaptation to constantly changing environmental conditions [49].

## 4. Candida glabrata

*C. glabrata* is a haploid budding yeast normally found as part of the microflora of healthy human mucosal tissues. Similar to *C. albicans,* it is a commensal colonizer that can turn opportunistic pathogen in immunocompromised patients. In the United States, *C. glabrata* is the second most common cause of candidiasis in humans, and it is the second most frequent causative agent of mucosal and bloodstream fungal infections [1,50,51,52]. In Europe and Asia, *C. glabrata* is the third most common non-*Candida albicans Candida* [53]. It is responsible for a good percentage of the blood infections caused by *Candida* spp., with more than a doubling of cases due to *C. glabrata* infections from 1992, reaching 24–26% in 2004–2008 in the United States [1,50,54], and accounting for over 12% bloodstream infections worldwide [51,55,56]. *C. glabrata* has emerged as an important nosocomial fungal pathogen associated with a high mortality rate, around 40%, comparable to the ones associated with *C. albicans* [57]. A likely contributor to this is increased resistance to antifungal agents, particularly azoles [50,54,55,58,59]. In addition, the prevalence of implant infections caused by fungal pathogens has dramatically increased over the last 10 years. Implanted devices, such as catheters, provide pathogens with surfaces on which they can adhere and form biofilms and *C. glabrata* has also been shown to form biofilm on a range of plastic surfaces [60].

*C. glabrata* telomerase is composed by cgEst2 catalytic subunit that is an orthologue of ScEst2 from *S. cerevisiae.* The RNA component TLC1, however, shows an internal inversion making it the largest telomerase RNA subunit with a length of over 2 kb. The RNA subunit shows the basic functional elements found in other fungi, but it also shows high sequence divergence due to large insertions. Loss of telomerase activity in *C. glabrata* results in shortening of the telomeres and senescence in most cells, but also gives rise to survivors. The survivors show heterogenous telomeres length, possibly due to homologous recombination. The event, however, does not seem to affect the sub-telomeric regions and possibly relies on a mechanism observed in *K. lactis* where a long telomere could suppress senescence through non-reciprocal recombination [61]. More research is needed to determine if the length and variability of the telomerase RNA component gives *C. glabrata* an advantage during a pathogenic switch.

Although not much is known about the factors that facilitate *C. glabrata* virulence and pathogenicity, like for other fungi, it is believed that the ability to strongly adhere to host cells and other surfaces is a critical contributing factor. In vitro, *C. glabrata* adherence to epithelial cells is mediated by a large adhesin, Epa1, which is generally expressed in vitro during lag phase and is repressed during log and stationary phase [62,63]. Epa1 belongs to a broad class of glycosylphosphatidylinositol-anchored cell wall proteins (GPI-CWPs) found in other species =, such as *S. cerevisiae*, *C. albicans*, and *A. fumigatus* (Table 4). Various studies showed that *EPA1* is part of a larger family of adhesin genes in *C. glabrata,* and that, like for other pathogens, including *P. jirovecii,* the cluster of genes shows a sub-telomeric localization and it is regulated through sub-telomeric silencing mechanisms [60,64,65]. De Las Peñas et al., [64] characterized *EPA2*-*EPA5* with the genes divided in two clusters, *EPA1*-*EPA3* and *EPA4*-*EPA5*, found at sub-telomeric regions. Deletion of the two different clusters of genes shows a moderate decrease in colonization, specifically of the kidneys, but altogether only showed a small effect on the yeast virulence. Interestingly, however, the only gene expressed in the various in vitro conditions was *EPA1*, indicating that the rest of the members are generally silenced. Hyper-adherent mutants allowed for the isolation of two novel *EPA* genes, *EPA6* and *EPA7* [65]. The two adhesins were also found while exploring factors involved in *C. glabrata* biofilm formation [60]. In all cases, the regulation of expression of the genes was found to be through the sub-telomeric silencing machinery.

Phylogenetically, *C. glabrata* is more closely related to *S. cerevisiae* than to *C. albicans*, as close homologues of *S. cerevisiae* genes can be found in the *C. glabrata* genome and both genomes conserve a high degree of synteny [68,69]. Considering the reported synteny between genes in the two organisms, it has been explored if similar sub-telomeric silencing complexes were acting in both organisms. The silencing of the adhesin genes is, in fact, mediated by proteins similar to the machinery used for telomeric silencing in *S. cerevisiae;* however, interestingly enough, the family cluster of adhesin genes are not found in *S. cerevisiae* [64] and the highly homologous silencing complex proteins seem to have different roles in the two organisms [60]. *C. glabrata* has in effect homologues of all the *S. cerevisiae* genes implicated in transcriptional silencing at the telomeres barring *SIR1*. Silencing in *S. cerevisiae* is mediated by binding of RAP1 to the telomeric repeats. The bound protein is known to recruit the SIR complex. One of the components of the SIR family, the histone deacetylase SIR2, which targets H3 and H4, primes the region increasing the binding of two other components, SIR3 and SIR4 and extending the transcriptional repression (Table 4). Another sets of proteins RAP1 interacts with are RIF1 and RIF2, which are involved in telomere length regulation, with RIF1 having a role in the silencing of both the telomeric and sub-telomeric regions (Table 4). Both *sir3Δ* strains and strains with a mutated *RAP1* de-repressed the transcription of the *EPA2-5* genes, albeit silencing differed for the two different clusters with a complete silencing for *EPA4/5,* and a more gradual telomere-distance dependent silencing for the *EPA1-4* cluster [64]. In the *C. glabrata* hyper-adherent mutants increased adherence was also affected by *EPA* genes, and *RIF1* and *SIR3*. *sir3*Δ and *rif1*Δ strains showed induction of genes from the whole *EPA* family, with *EAP1*, *EPA6,* and *EPA7* being the main, albeit not the only, drivers of the hyper-adherent phenotype. *RIF1*, *RAP1,* and *SIR3* are required in *C. glabrata* for the silencing of genes at sub-telomeric locations with silencing generally decreasing as the distance from the telomere increases.

Overexpression of *EPA1*, *EPA6,* and *EPA7* in *sir3*Δ strains has been shown to be the cause of *C. glabrata* hypercolonization of the kidneys, implicating SIR-mediated silencing in the regulation of the *EPA* genes expression and in *C. glabrata* virulence. Analysis of *EPA1* expression in individual cells showed that the expression is highly heterogenous and it is controlled at the transcriptional level, and that the mode of silencing is also cell-specific, with some cells depending on SIR, and others relying on some other transcriptional mechanism [70]. These types of behavior are believed to be connected to genes that respond to environmental changes. For example, the lack of nicotinic acid can affect SIR2, a NAD^+^-dependent histone deacetylase, activity that in turn would derepress *EPA1* expression. This could be a drive in kidney infection [71]. *C. glabrata* contains four more gene homologues of *SIR2: HST1*–*HST4*. The closest paralogue of *SIR2* is *HST1,* which, like *SIR2* is involved in transcriptional repression, but its function does not overlap with *SIR2. HST1* is also dependent on niacin to work. Loss of *HST1* increases EPA6 transcription and adherence of *C. glabrata* to epithelial cells, and increases *C. glabrata* resistance to fluconazole and hydrogen peroxide by derepressing the silencing of other genes [71,72]. These findings suggest a model in which the pathogen in a specific environment, e.g., the genitourinary tract, can increase its ability to colonize the organs and acquire multidrug resistance. Furthermore, it can develop better oxidative stress response.

In addition to sub-telomeric silencing, *EPA1* expression is negatively regulated by a *cis*-acting negative element. This element is a 200-bp sequence located in the intergenic region between *EPA1* and *EPA2*. It is 300-bp downstream from the *EPA1* stop codon and is the binding site for yKu70 and yKu80. Hence, *EPA1* expression is controlled by sub-telomeric silencing through the SIR complex and by a telomere independent mechanism involving the *cis*-element in association with yKu70/80 [63]. *EPA1* expression has also been shown to be under the influence of different genes on different telomeres. Juárez-Reyes et al. [73] examined at the only telomere among the ones they analyzed that was not under the control of yKu70 and yKu80 for *EPA* silencing, and found a proto-silencer, Sil2126, located between the *EPA3* gene and the telomere. It has an overlapping function with yKu70 and yKu80 so that E_-R_ telomere seems to be independent of yKu70 and yKu80 control [67]. Considering the telomere apparent independence from yKu70 and yKu80 when Sil2126 is present and the pro-silencer limitations, it is fair to assume that other *cis*-acting element may be present at the sub-telomeric region to allow Sil2126 activity. Silencing, in case of Sil2126 loss, is mediated by yKu70 and yKu80 [73]. Proto-silencer Sil2126 contains binding sites for RAP1 and ABF1, both known to mediate the formation of heterochromatin through their recruitment of the SIR complex. Sil2126 can strongly interact with an intergenic region between *EPA2* and *EPA3* and create a loop. The proto-silencer can also interact with the negative *cis*-element found between *EPA1* and *EPA2*. Through these interactions and the recruitment of the various proteins responsible for sub-telomeric regions silencing, it is believed that expression of *EPA1*, *EPA2*, and *EPA3* is kept under strict control by the cell. It is possible that *cis*-elements are present, particularly at other telomeres, since the mechanisms seem to be different on different chromosomes [74]. The phenotype for the *rif1*Δ, which showed a decrease of liver and spleen colonization, seems to be more complex. The increase of telomere length due to *RIF1* deletion may have broader effects than the loss of sub-telomeric silencing seen with *sir3*Δ [65]. It is hypothesized that the increase in telomere length in the *rif1*Δ strain titrate SIR proteins from other chromosomal sites, reducing silencing at those sites and concomitantly increasing silencing of genes at more telomeric locations and confirming RIF1 role in telomere length in *C. glabrata* [65].

In analyses of biofilms, it has been showed that the known adhesins do not seem to have a role in *C. glabrata* cells adhesion to plastic substrates, but both *EPA6* and *EPA7* are expressed after biofilm establishment with *EPA6* being one of the main players for cell-to-cell adhesion in the formed biofilm. It is possible other adhesins in the family mediate the initial stages. The sub-telomeric silencing control of these genes in biofilms was confirmed by looking at *sir4*Δ and *sir4*Δ epa6Δ mutant strains. Both strains showed the same level of biofilm formation, suggesting that release of silencing of the gene cluster may be more important than just *EPA6* expression. It was in fact observed both in *sir4*Δ and *rif1*Δ that *EPA1*-*EPA5* and *EPA7* were overexpressed [60].

The silencing and release of repression are the basis of the different expression of the adhesin family members in the various mutant backgrounds. These results suggest that their transcription is likely mediated by the sub-telomeric silencing machinery in response or in addition to responses to particular environmental clues. Rosas-Hernandez et al., [67] looked at the role of the proteins in the various telomeric complex for specific telomeres in *C. glabrata* and found that silencing of different sub-telomeric regions is based on different requirements. They established that *HDF1* and *HDF2,* coding for yKu70/80, are functional equivalents of the *S. cerevisiae* genes and have a role in telomere maintenance and non-homologous end join repair (NHEJ) (Table 4). They also found that although Rap1, Sir2, Sir3, and Sir4 are required for sub-telomeric genes silencing at various telomeres, Rif1, yKu70/80 function has a different impact at different sub-telomeric regions. In *C. glabrata,* the sub-telomeric regions subjected to TPE can reach up to 25 kb, much larger than in other known fungi, and *SIR2*, *SIR3*, and *SIR4* are required for TPE to work. Rif1, yKu70, and yKu80, however, do not have such blanket effect. yKu70 and yKu80 are required for TPE at some chromosomes, but not all. Rif1 has a positive role in telomeres silencing but its effect varies at different telomeres and shows discontinuous silencing across the telomere. The differential effect of these proteins on the telomeres may be due to the presence of *cis*-acting silencers at sub-telomeric regions. A silencer regulating *EPA3* was isolated in *C. glabrata* with its function based on interactions with Sir3. Its function, based on data from silencers in *S. cerevisiae*, is correlated to telomeres length and consequent availability of Sir3. Longer telomeres, like in *rif1*Δ, would mean more Rap1 bound to the DNA and greater recruitment of the SIR complex to telomeres and less to the *cis*-element and hence de-repression of the gene expression. Shorter telomeres, like in *hdf1*Δ and *hdf*2Δ, and more availability of Sir3, results in stronger silencing of the sub-telomeric genes at that region [67]. The complex regulation of expression of adhesin genes with differential silencing and expression of the proteins at different telomeres in response to environmental clues can be the basis of the antigen heterogenicity on different cells in the same population and guarantee an escape mechanism for the pathogen during infections in vivo.

yKu70, yKu80, and Rif1 are also involved in sub-telomeric silencing of the *MTL3* locus in *C. glabrata.* Although *C. glabrata* has never been shown to mate and in general human pathogenic fungi do not show a sexual cycle, it has been hypothesized that mating could activate specific combinations of genes required for virulence and survival in the host. The *MTL3* silencing is imperfect and can potentially have a role in cell-specific genes seen in *C. glabrata* that could be correlated with pathogenicity and virulence [75].

Genomic rearrangements in *C. glabrata* have been observed in clinical specimens and seem to be correlated to the fungus virulence and drug resistance [76]. This type of behavior leading to genomic instability, including the creation of new chromosomes, is generally deadly, but can be observed in haploid organisms that rely on asexual reproduction. Albeit, such rearrangements have been seen in other yeasts, such as *S. cerevisiae*, to which *C. glabrata* is evolutionarily closer, it is usually observed under lab conditions and as a result of mutant background. *C. glabrata* shows a high rate of non-reciprocal translocation, chromosome fusion, novel chromosomes giving rise to varying karyotypes. Such variations have been observed in other pathogens in connection with increase in drug resistance and virulence. It has been hypothesized that such dramatic genomic changes in *C. glabrata* are due to telomeric instability. In fact, both *TEN1* and *RIF2*, which are important for telomere maintenance, are missing in the pathogen and known in *S. cerevisiae* to be involved in telomere end protection and length regulation [25].

## 5. Pneumocystis Fungi

*Pneumocystis* are parasitic fungi that become pathogenic upon weakening of the immune system of the host, causing pneumonia that can turn fatal. The organism is often associated with the AIDS pandemic as it was one of the first diseases associated with it. In the 1980s, Pneumocystis infections were found in half of the adult AIDS patients and were associated with high mortality. The numbers of infections in developed countries have decreased, but the pathogen is still one of the deadliest worldwide. Even in countries where modern drugs have helped in reducing the more dire effects of HIV infections, *Pneumocystis* infections are still a critical problem in patients undergoing immunosuppressive treatment for other chronic medical issues. In fact, *Pneumocystis* pneumonia has been observed to have a higher incidence of death in non-AIDS patients, 40–70%, than in AIDS-patients, 10–30% [77].

The understanding of *Pneumocystis* biology has been hindered by the difficulties to cultivate and propagate the organism in vitro. *Pneumocystis* is an obligate parasite and comparative genomic studies show that the evolution of *Pneumocystis jirovecii* involved the loss of genes from pathways essential for independent growth and reproduction [78]. Due to this situation, much of the data regarding the organisms come from samples taken from *Pneumocystis* infected humans or rats or from comparative genomic analysis [79,80,81,82,83]. The obliged reliance on the host may be the reason why different pneumocystis strains generally target mammalian lungs in a very species-specific way.

Data on the telomere structure and associated proteins of *Pneumocystis* are limited. What is known is that *Pneumocystis jirovecii* has telomeres composed of 5′-TTAGGG-3′ repeat sequences and that telomeres should encompass approximatively 10% of the genome [3,43,84] Some of the telomeric repeats have also been observed at non-telomeric sites, possibly as the result of chromosome rearrangements or as internal repeats part of sub-telomeric sequences [85,86]. Similar to other fungi, two of the key components of *Pneumocystis* virulence are its ability to adhere to the epithelial cells in the alveoli of the host and the ability to offset the response of the host immune system. Although, due to the technical difficulties in culturing the organism, the players involved in cell adhesion have not been well characterized. A superfamily of surface proteins composed by eight families of genes involved in antigenic variation were discovered in *Pneumocystis* and they are specific to the organisms. All but one are composed by clusters localized at sub-telomeric regions [81,87].

The diversity in the genes within the surface proteins family and among family may be due to the localization that could favor genetic recombination between members of each gene family over time. Schmid-Siegert et al. [82] observed high variability of the sub-telomeres between *P. jirovecii* isolates, which has been associated with frequent sub-telomeric recombination. Pseudogenes are also be maintained within the sub-telomeric regions possibly so the organism can keep a reservoir to use for the creation of mosaic genes. Telomeric genes can also undergo mitotic recombination without causing major genomes rearrangements and can involve non-homologous chromosomes. A sub-telomeric localization for these family clusters may support mutually exclusive expression through gene silencing observed in one of the main families.

Cell wall composition in fungi varies depending on the life cycle stage and may also be relevant for the fungus pathogenicity and virulence. *Pneumocystis* has been known to present different life cycle states such as precystic, cystic, and trophic, with the trophic form being predominant during active infection [77]. Some members of the major surface glycoproteins (MSG), the most abundant components of the cell wall and a key target for the host immune defense, are known to interact with proteins in the extracellular matrix, like fibronectin, suggesting a role in cell adhesion; however, data show that their expression is consistent throughout different phases of life of the pathogen. A more relevant role for the *MSG* gene family seems to be to mediate the response of the host immune system to the parasite. It is believed that antigenic variation or mosaicism of the *MSG* expression is one of the mechanisms used by the fungus to evade their host defenses (Table 5; [81,83]. *MSG* family genes are clustered around the telomeres with 2 to 4 genes located at each of the end of the *Pneumocystis* chromosomes [80]. The family comprises approximatively 80 isoforms, but only a single isoform is expressed on the cell surface at any time while all the other members are transcriptionally silenced in what is called mutually exclusive expression. Interestingly, albeit every individual fungus expresses a single MSG isoform, individuals in the population can express multiple isoforms during infection as observed in *Pneumocystis jirovecii* [88]. MSG isoforms are also expressed differently, and may differ in different species mirroring the species specificity of the organism [88]. The host specific expression may facilitate evasion of the host immune system [88].

Using long read sequencing technology, the highly repetitive sequences of the *Pneumocystis* genome, specifically for *P. jirovecii* and *P. murina*, has been characterized [81]. These sequences encompassed the large gene families of MSG and kexin proteases. In the reduced *Pneumocystis* genome, the *MSG* gene family represents 3–6% of the genetic material underlying its importance for the organism. Both families are found in sub-telomeric regions where most of the genetic variation was found in *P. jirovecii*.

It has also been found that each *Pneumocystis* genome encodes approximately 60–180 *MSG* genes, depending on the species, including the classical *MSG* genes, *MSG*-related (*MSR*) genes, and additional related genes [81]. These genes are collectively termed the *MSG* superfamily. Based on domain structure, phylogeny analysis and expression control mechanisms of the msg superfamily, it has been proposed a classification of five families, named as *MSG-A*, *MSG-B*, *MSG-C*, *MSG-D*, and *MSG-E* (Table 6; [88]). According to the chromosomal-level assemblies of the *P. murina* and *P. jirovecii* genomes, *MSG* genes are located almost exclusively in subtelomeric regions and usually present in clusters. Expression of the *MSG* genes is under the control of a promoter region called the upstream conserved sequence (*UCS*). The sequence is located upstream of the individual *MSG* gene expressed by the organisms. The *UCS* includes both promoter and the leader sequence that is responsible for the protein translocation to the endoplasmic reticulum for the incorporation into the cell wall. The mechanism used by *Pneumocystis* to switch between isoforms is still unclear, but it is hypothesized that the telomeric *UCS* location may promote recombination events that allow switching *MSG* gene expression [89]. More specifically, site-specific recombination involves *USC* and a sequence found at the beginning of each of the *MSG* termed conserved recombination junction element (*CRJE*). Members of the family can be expressed only upon recombination of their *CRJE* with that of the single-copy *UCS*. Schmid-Siegert et al., [82] suggested that mosaic genes could be created by homologous recombination within each *MSG* family. They hypothesized that genes closest to the telomeres may undergo recombination at the *CRJE* sequences for the exchange of the single gene and that this may facilitate telomere exchanges [82]. In addition, the *CRJE* sequence has a recognition site for Kex1, the protease coded by the *PRT1* gene also found at sub-telomeric regions, which may be involved in the maturation of the MSG antigens and potential add another layer of complexity to the expression of mosaic genes.

Two other characterized gene families, *MSR* genes family and the *PRT1* family, located at telomeres in *Pneumocystis jirovecii,* may also be involved in immune evasion by increasing surface antigenic variation (Table 6; [77]). Their expression does not seem to be dependent on the *UCS* and the organism will express a variety of these protein isoforms. While MSR, like MSG, proteins are found on the cell surface, the Ptr1 polypeptide that has been described in *P. jirovecii* codes for a protease. In *P. jirovecii*, the proteases are located on the cell surface and it is hypothesized that they are involved in modification of cell surface proteins on location and hence in increasing antigenic variation. Another hypothesis is that they have a role in degrading host protein affecting pathogen survival. The *PRT1* isoform characterized in *P. jirovecii* codes for Kex1, a protease that is believed to have a function in the Golgi apparatus, possibly processing MSG [77]. Keely et al., [80] sequenced six telomeric segments and found the members of the gene families were generally structured in *PTR1*-*MSR*-*MSG* repeats, suggesting the possibility of a readthrough transcription [80]. However, it is not likely the case as the various *PTR1* and *MSR* family members can be consistently expressed, while as mentioned only one particular *MSG* is expressed at the time [79]. The mechanism of the variations in the gene expression in this family is not known, but their telomeric/sub-telomeric location may also be instrumental. Genes that are not located at the expression site can undergo telomeric silencing. In addition, sequences of intergenic spacer were found to be more conserved than the genes themselves and were present both at telomeric and sub-telomeric regions. These spacers have been known to have a role in gene expression, by silencing adjacent genes or impeding the spread of repressive chromatin, and recombination by regulating recombination of telomeric genes through the association of chromosome ends.

## 6. Conclusions

Telomeres, the long non-coding sequences found at the ends of linear DNA, have long been associated with DNA replication protection of chromosomal ends [90]. In this review, however, we demonstrated that the value of these sequences, and more importantly their sub-telomeric counterparts, goes far beyond just a protective mechanism in various fungal species. In many opportunistic fungal pathogens, the virulence and adaptability of those species is found influenced by the expansion of highly variable sub-telomeric sequences flanking telomeres [91]. Gene clusters yielding secondary metabolites influencing virulence, as well as hyphal extension abilities for added adherence and pathogenicity, are amongst the sub-telomeric components identified in opportunistic pathogen activities in a variety of fungi, including *Candida* and *Aspergillus* species that are amongst the most problematic for humans [27,41]. Gene families in telomeres of *Pneumocystis* have even demonstrated involvement in cell surface antigens [77]. Thus, not only are telomeric and sub-telomeric sequences of potentially critical value in understanding the virulence, adherence, and adaptability of fungal pathogens, but also the immune responses and evasive activities as well. It is evident that increased attention to studying the telomere and sub-telomeric structures and functions of fungal species could have great implications in the medical field, as potential targets of treatment and control of pathogens.

Although telomeres have an essential role in the survival of an organism in many species, including yeasts, both telomeric sequences and telomeric proteins have been undergoing evolutionary changes. Most fungi retain the canonical sequence. However, the model organisms *S. cerevisiae* and *Schizosaccharomyces pombe,* similar to other species, such as *C. albicans, Kluyveromyces lactis*, show variation both in sequence and in length of the telomeric repeats, with *C. albicans* and *K. lactis* reaching over 20 nucleotides in length [92]. Similarly, yeast telomeric proteins span from showing similarities to the mammalian ones, such as in *S. pombe,* to distinct and independent DNA binding proteins, such as Rap1 in *S. cerevisiae* or like Tbf1, also in *S. cerevisiae,* having a role also in subtelomeres regulation and structure and as a transcription factor elsewhere in the genome [93,94]. The co-evolution of telomeric sequences along with their accompanying proteins and chromatin has an important role in fungi to survive and become pathogenic, but there is still much to learn about the topic. Epigenetics and chromatin structure play a role in telomeric and subtelomeric influences on pathogenicity, with recent studies demonstrating heterochromatin’s repressive effects on gene expression modifies gene stability and thus pathogenicity in variable environments of common human fungi, but much more research in this area needs to be performed [95]. Furthermore, it should be noted that not only has pathogenicity in fungi been found influenced by these epigenetic and telomeric regulations, but also various other Eukaryotic pathogens, such as protozoan parasites, appear to be subject to similar mechanisms [96]. Overall, epigenetics and genomics open a whole new world of information on pathogenicity and infectious disease understanding/targeting.

## Figures and Tables

**Figure 1 microorganisms-09-01405-f001:**
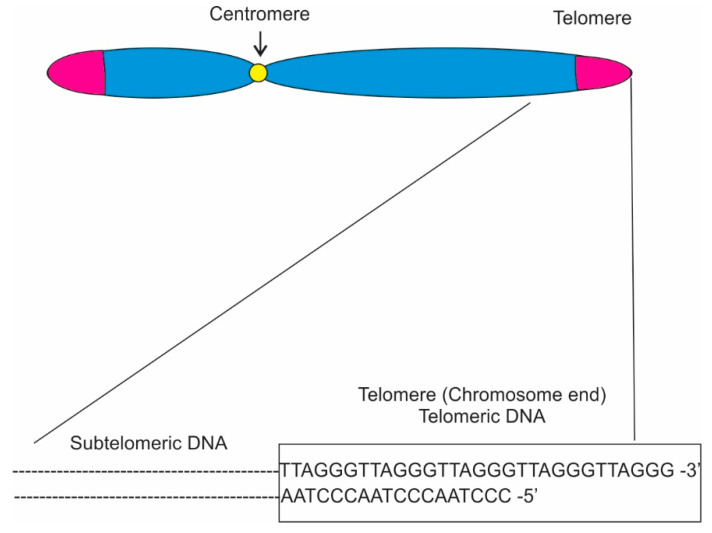
The DNA sequences of a typical chromosomal end structure.

**Figure 2 microorganisms-09-01405-f002:**
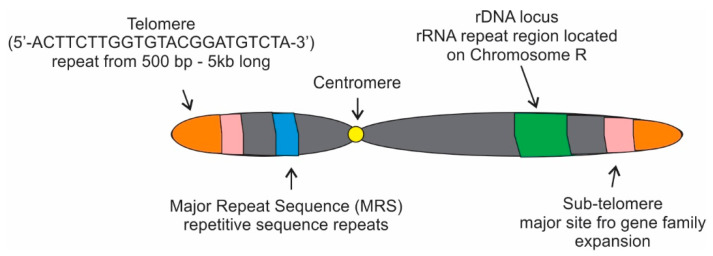
Major classes of genetic repeats in *Candida albicans*. *C. albicans* contains four major categories of repeat sequences: the telomeres that contain multiple copies of a 23 bp repeat; the major repeat sequence (MRS) composed of repetitive sequence repeats; the rDNA locus, which encodes the polycistronic rRNA transcripts; and the sub-telomeres, a telomere-proximal region containing transposable elements and a gene family expansion.

**Table 1 microorganisms-09-01405-t001:** Telomere repeat sequences and base pair length for all four opportunistic fungi of this review.

Fungal Species	Telomere Repeat Sequence	Number of bp Repeats
*Aspergillus fumigatus*	5′-TTAGGG-3′	6
*Candida albicans*	5′-ACTTCTTGGTGTACGGATGTCTA-3′	23
*Candida glabrata*	5′-CTGGGTGCTGTGGGGT-3′	16
*Pneumocystis jirovecii*	5′-TTAGGG-3′	6

**Table 2 microorganisms-09-01405-t002:** Subtelomeric gene cluster members that increase pathogenicity in *A. fumigatus*.

Gene/Genes in Subtelomeres	Protein/Metabolite	Function	References
Fumitremorgin biosynthesis supercluster	Pseurotin	Neuritogenic, nematicidal quinone	[28,29]
*LaeA* gene	LaeA methyltransferase	Global transcriptional regulator of secondary metabolite biosynthesis; virulence factor	[30]
*bafA*	Biofilm Architecture Factor A	Modulates hyphae formation and ability to survive low oxygen conditions increasing pathogenicity	[31]
*atmA* and *atrA*	Ataxia-telangiectasia mutated kinases	Chromosomal polymorphism	[32]

**Table 3 microorganisms-09-01405-t003:** Subtelomeric genes that increase pathogenicity in *C. albicans*.

Gene/Genes	Protein/Metabolite	Function	References
*TER1*	Telomerase RNA component	Involved in production and maintenance of telomeres to prevent cell senescence	[40,44]
*EST2*	Telomerase reverse transcriptase	Involved in production and maintenance of telomeres to prevent cell senescence	[40,44]
*EST3*	Subunit of telomerase regulatory complex	Telomerase stimulatory function	[45,46]
*RAP1*	Repressor activator protein 1	Regulator of telomere length and structure; works in conjunction with Ku70, Stn1, and Ten1	[47]
*TLO* genes	Highly variable Telomere associated metabolites	Includes Med2 and other subunit components of mediator complex and transcriptional responses; Involved in adaptability, hyphal growth, and oxidative stress responses	[41,48]

**Table 4 microorganisms-09-01405-t004:** Subtelomeric genes that increase pathogenicity in *C. glabrata*.

Gene/Genes	Protein/Metabolite	Function	References
*EPA1–7*	Large adhesin proteins (glycosylphosphatidylinositol-anchored cell wall protein)	Adherence; clusters of subtelomeric *EPA* genes are involved in colonization and biofilm formation	[62,63]
*SIR 1–4*	Histone deacetylase and silencing proteins	Involved in transcriptional repression, gene silencing, and adhesion	[64]
*RAP1*	Repressor activator protein 1	Regulator of telomere length and structure; Involved in gene silencing	[64]
*RIF1–2*	Rif1 and Rif2 regulatory Proteins	Regulators of telomere length and structure; Involved in gene silencing	[66]
*HDF1–2*	yKu70/80	Telomere maintenance; non-homologous end join repair	[67]

**Table 5 microorganisms-09-01405-t005:** Subtelomeric genes that increase pathogenicity in *Pneumocystis*.

Gene/Genes	Protein/Metabolite	Function	References
*MSG*	Major Surface Glycoproteins	Components of cell wall and involved in antigenic variation evading host immune responses	[81,83]
*MSR*	MSG-related antigens	Part of the MSG superfamily involved in isoform switching and immune evasion with antigenic variation	[77]
*PRT1*	Cell surface protease	Involved with MSR in immune evasion with antigenic variation	[77]

**Table 6 microorganisms-09-01405-t006:** Summary of the MSG superfamily members identified in *Pneumocystis* species.

Family	No. of MSG Genes			Characteristics			
	*P. murina*	*P. jirovecii*	Average Gene (kb)/Protein (kDa) Size	No. of Introns	5′-end Leader Sequence	No. of Domains	Expression Mode
*MSG-A1c*	22	86	3.2/120	0	*CRJEd*	9	Mutually exclusive	
*MSG-A2c*	14	0	2.9/218	1	Highly conserved	7–9	Independently	
*MSG-A3c*	6	33	3.1/117	1–2	Highly variable	9	Independently	
*MSG-B*	0	21	1.4/55	1–2	Highly variable	2–3	Independently	
*MSG-C*	6	2	1.7/60	1	Highly conserved	3	Independently	
*MSG-D*	1	20	2.9/111	1	Highly variable	3–6	Independently	
*MSG-E*	7	5	1.2/49	1	Highly variable	1	Independently	
Unclassified	8	12	0.9/37	0–8	Highly variable	0–1	Unknown	

## Data Availability

Not applicable.

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
