# Peer review of "Telomeric and Sub-Telomeric Structure and Implications in Fungal Opportunistic Pathogens"

_microorganisms, 2021, doi:10.3390/microorganisms9071405_

Round 1

Reviewer 1 Report

The review by Diotti, Esposito, and Shen summarises molecular and genetic work around telomere maintenance and sub-telomeric gene clusters in four species of clinically important opportunistic pathogen fungi. The review is well written and includes highly detailed study summaries, as well as extended discussions of clincal importance and evolutionary context. I would recommend this manuscript for publication. 

Minor comments:

There were two sentences with a bit unclear message for me: at lines 50-53, and at lines 151-153. 

Line 144 - and TO shed light 

Not mentioned in review, but might have some relevance: subtelomeres of parasitic apicomplexans (e.g., Plasmodium) and kinetoplastids (e.g., Trypanosoma) also have similar associations with environment interaction (immunity evasion, drug resistance etc). 

Author Response

Reviewer 1:

The review by Diotti, Esposito, and Shen summarises molecular and genetic work around telomere maintenance and sub-telomeric gene clusters in four species of clinically important opportunistic pathogen fungi. The review is well written and includes highly detailed study summaries, as well as extended discussions of clincal importance and evolutionary context. I would recommend this manuscript for publication. 

Minor comments:

There were two sentences with a bit unclear message for me: at lines 50-53, and at lines 151-153. 

-Thank you, we have reworded both sentences to correct and clarify the meanings.

Line 144 - and TO shed light

Thank you, we have corrected the grammatical error. 

Not mentioned in review, but might have some relevance: subtelomeres of parasitic apicomplexans (e.g., Plasmodium) and kinetoplastids (e.g., Trypanosoma) also have similar associations with environment interaction (immunity evasion, drug resistance etc). 

-To address this, we modified our closing remarks to demonstrate the value of these mechanisms beyond just fungi and towards other Eukaryotic pathogens, including the protozoan parasites as you mentioned.

Reviewer 2 Report

This review describes the importance of the telomeric and sub-telomeric structures in opportunistic fungal pathogens (A. fumigatus, C. albicans, C. glabrata, and P. jiroveci), interns of pathogenicity, virulence, abilities and biodiversity.

The work is very well written and organized, and the topic is opportune and interesting. Antifungal resistance is indeed growing fast, and novel therapeutic responses are urgently needed.

The major point this reviewer finds is that the review has many information and tables or schemes should be performed since they would ease the comprehension of the telomeric/sub-telomeric/genes discussed. Also, the references should be more recent.

Other points are:

Introduction:

- References need to be adjusted in the format indicated by the journal;

- Line 42: “ Aspergillus spp” – correct “spp” to “spp.”;

- There has been some confusion with Pneumocystis jirovecii and Pneumocystis carinii. Sometimes they’re used as the same. Can the authors please clarify which species they’ve approached in this work?

- Line 97: “S. cerevisiae”: please write the entire name of any species the first time is mentioned;

- Line 162: “virulence in a murine model. (Perrin et al., 2007).” –  please remove “.” before the reference;

- Line 199: Yang et al. (2003) – this reference is old should be replaced by other more recent work(s). There are several;

- Line 199, 219: “Candida species” – replace by Candida spp.;

- Tables related to telomeric/sub-telomeric genes/protein/action, role, or pathogenicity relation(/reference(s)) of each pathogen would be important. It could really help understanding what is explained in each section;

- Lines 305-307: “ C. glabrata is the second most common cause of candidiasis in humans, and it is the second most frequent causative agent of mucosal and bloodstream fungal infections...”. - This sentence is not accurate. C. glabrata is, indeed, the second most common Candida in the United States, but it is third in Europe (and Asia). Please check: https://doi.org/10.3390/jof3010011; doi: 10.1371/journal.pone.0252165;

- Line 336: “EPA1” is this a protein or a gene? If it is a protein it need to be as “Epa”; If it is a gene, it requires italic form. There are other cases in the MS. Please check the entire MS and correct this detail;

- This reviewer thinks there are several old references. Many works have been published in the last 5 years related to Candida, Aspergillus, resistance and virulence and should be cited, replacing the oldest ones;

- Line 493: italic form is missing on “Pneumocystis”;

- Line 635: “S. pombe” – please define all abbreviations before using it.

Reviewer 3 Report

The authors reviewed the organization of telomere and sub-telomere and their relations with virulence in four opportunistic fungi.  

- The authors have chosen to describe separately the four pathogens in distinct paragraphs. This choice has some pertinence but it leads to some repetition about the structure of the shelterin complex for instance. An additional paragraph could have been added, after introduction, in order to describe more generally the silencing mechanisms and the organization of proteins bound to telomere in fungi. The manuscript would be then easier to follow for the reader. Also, each paragraph could be divided in subsections with explicit titles.

- T-loops and loopback in yeast (line 239) are still subject to debate (for instance, Pasquier & Wellinger doi: 10.1186/s13072-020-00344-w.)

- There are many problems with references: some are missing, misformatedor duplicated.

Missing:

Saint-Leandre & Levine 2020

Nierman et al 2019

Kowalski et al 2021

Misformated

Mceachern & Blacburn 1994

Brocolli et al 1997 (missing volume and pages)

McDonagh et al 2008 (duplication of authors)

Duplicated

De Lange 2005

Palmer & Keller 2011

- Misspelling

line 210 : « Legrande et al, 2019 » (Legrand et al, 2019)

line 606 : are not at located at the expression

Author Response

The authors reviewed the organization of telomere and sub-telomere and their relations with virulence in four opportunistic fungi.  

- The authors have chosen to describe separately the four pathogens in distinct paragraphs. This choice has some pertinence but it leads to some repetition about the structure of the shelterin complex for instance. An additional paragraph could have been added, after introduction, in order to describe more generally the silencing mechanisms and the organization of proteins bound to telomere in fungi. The manuscript would be then easier to follow for the reader. Also, each paragraph could be divided in subsections with explicit titles.

-Thank you so much for your valuable feedback. We have revised the manuscript and added 4 new tables to help make the manuscript easier to follow for the reader. We have also added the suggested additional paragraph after the initial introductions.

- T-loops and loopback in yeast (line 239) are still subject to debate (for instance, Pasquier & Wellinger doi: 10.1186/s13072-020-00344-w.)

-To take into account the loop debate, we have now specified “in some fungal species” in that sentence to demonstrate that the pathway/mechanism described for the particular species of that paragraph does not represent a definitive mechanism of all fungi.

- There are many problems with references: some are missing, misformatedor duplicated.

-Revised entire reference list to conform to journal guidelines and corrected each of the following discrepencies:

Missing:

Saint-Leandre & Levine 2020

Nierman et al 2019

Kowalski et al 2021

 -Corrected

Misformated

Mceachern & Blacburn 1994

Brocolli et al 1997 (missing volume and pages)

McDonagh et al 2008 (duplication of authors)

 -Corrected

Duplicated

De Lange 2005

Palmer & Keller 2011

 -Corrected

- Misspelling

line 210 : « Legrande et al, 2019 » (Legrand et al, 2019)

-Corrected

line 606 : are not at located at the expression

-Corrected grammatical error

Round 2

Reviewer 2 Report

Dear Authors,

Thank you for the reply and the adjustments.

Good luck for future works.